# Mobile Phone Use and Mental Health. A Review of the Research That Takes a Psychological Perspective on Exposure

**DOI:** 10.3390/ijerph15122692

**Published:** 2018-11-29

**Authors:** Sara Thomée

**Affiliations:** Department of Psychology, University of Gothenburg, 405 30 Gothenburg, Sweden; sara.thomee@psy.gu.se; Tel.: +46-31-7861882

**Keywords:** cell phone, epidemiology, psychology, behavioral addiction, depression, sleep

## Abstract

The purpose of this study was to carry out a review of observational studies that consider links between mobile phone use and mental health from a psychological or behavioral perspective. Systematic literature searches in PubMed and PsycINFO for articles published until 2017 were done. Exclusion criteria included: papers that considered radiofrequency fields, attention, safety, relational consequences, sexual behavior, cyberbullying, and reviews, qualitative, and case or experimental studies. A total of 4738 papers were screened by title and abstract, 404 were retrieved in full text, and 290 were included. Only 5% had any longitudinal design. Self-reporting was the dominating method of measurement. One third of the studies included children or youth. A majority of adult populations consisted of university students and/or self-selected participants. The main research results included associations between frequent mobile phone use and mental health outcomes, such as depressive symptoms and sleep problems. Mobile phone use at bedtime was associated with, e.g., shorter sleep duration and lower sleep quality. “Problematic use” (dependency) was associated with several negative outcomes. In conclusion, associations between mobile phone use and adverse mental health outcomes are found in studies that take a psychological or behavioral perspective on the exposure. However, more studies of high quality are needed in order to draw valid conclusions about the mechanisms and causal directions of associations.

## 1. Introduction

Mobile phones have over only a few decades revolutionized how we communicate, interact, search for information, work, do chores, and pass time. The development of the smartphone with its multitude of functions, increased memory capacity and speed, and constant connectedness to the internet, has increased the time spent using the phone, implying a near ubiquitous usage. This fast development with changed exposure patterns has raised questions about potential health effects of the exposure [1,2]. The mobile phone communicates through emission of radio signals, and the exposure to radiofrequency electromagnetic fields has been proposed to be a health risk. There are today few indications that radiofrequency electromagnetic fields associated with mobile phones have any major health effects [3]. The World Health Organization (WHO) is currently undertaking a health risk assessment of radiofrequency electromagnetic fields, to be published as a monograph in the Environmental Health Criteria Series [4]. However, in addition to physiological aspects of the exposure, there is a growing research literature that takes a psychological or behavioral perspective on potential health effects of mobile phone use. The purpose of this literature review was to supplement the work of the WHO expert group by carrying out a literature review of quantitative observational studies that consider links between mobile phone use and mental health from a psychological or behavioral perspective. A formal systematic critical review with quality assessment of the papers was not done due to the large amount of included studies. The report presents an overview of the studies and examples of the main results. It does not include a comprehensive account of all included papers.

## 2. Materials and Methods

Two skilled university librarians performed systematic literature searches in PubMed and PsycINFO on 2 May 2016, with supplemental searches on 19 March 2018. The final search strategies (Table 1) aimed to identify scientific publications from 1993 to 31 December 2017 that included quantitative analyses of mobile phone use in relation to mental health outcomes and other psychological factors. Altogether, 4738 papers were identified, after automatic removal of duplicates. These were screened by title and abstract. Papers that considered radiofrequency electromagnetic fields (RF-EMF), attention or safety (while driving, working, or studying), consequences for relationships, sexual behavior (e.g., sexting), cyberbullying, as well as papers that were qualitative, case or experimental studies, literature reviews, or duplicates (not previously identified), were excluded. This left 404 articles to be retrieved in full text for evaluation. Another 114 papers were removed in accordance with the previously mentioned exclusion criteria, or if no mental health-related outcome could be distinguished, if mobile phone use could not be identified as a separate variable (e.g., was included in a composite variable such as “digital media” or “screen time”), if focused only on specific smartphone applications (e.g., Tinder, Facebook, camera) or phone loss scenarios, or were not in English. This left 290 studies [5,6,7,8,9,10,11,12,13,14,15,16,17,18,19,20,21,22,23,24,25,26,27,28,29,30,31,32,33,34,35,36,37,38,39,40,41,42,43,44,45,46,47,48,49,50,51,52,53,54,55,56,57,58,59,60,61,62,63,64,65,66,67,68,69,70,71,72,73,74,75,76,77,78,79,80,81,82,83,84,85,86,87,88,89,90,91,92,93,94,95,96,97,98,99,100,101,102,103,104,105,106,107,108,109,110,111,112,113,114,115,116,117,118,119,120,121,122,123,124,125,126,127,128,129,130,131,132,133,134,135,136,137,138,139,140,141,142,143,144,145,146,147,148,149,150,151,152,153,154,155,156,157,158,159,160,161,162,163,164,165,166,167,168,169,170,171,172,173,174,175,176,177,178,179,180,181,182,183,184,185,186,187,188,189,190,191,192,193,194,195,196,197,198,199,200,201,202,203,204,205,206,207,208,209,210,211,212,213,214,215,216,217,218,219,220,221,222,223,224,225,226,227,228,229,230,231,232,233,234,235,236,237,238,239,240,241,242,243,244,245,246,247,248,249,250,251,252,253,254,255,256,257,258,259,260,261,262,263,264,265,266,267,268,269,270,271,272,273,274,275,276,277,278,279,280,281,282,283,284,285,286,287,288,289,290,291,292,293,294] for closer scrutiny (Appendix A. PRISMA Flow Chart). 

## 3. Results

The identified studies (*n* = 290) mainly dealt with frequency or duration of mobile phone use in relation to mental health symptoms (such as depression, anxiety, and insomnia), mobile phone use and sleep habits, and “problematic mobile phone use” (dependency/addiction). The number of published papers greatly increased during the time-period, especially the last five years (Table 2). 

### 3.1. Study Designs and Populations

A massive majority of the retrieved studies had cross-sectional design. Only 14 studies, i.e., about 5% [26,65,95,123,132,144,148,156,184,249,268,269,274,286], were identified as having any form of longitudinal design, test-retest reliability studies excepted.

About one third of the studies were based on child or adolescent populations, mostly administered through schools. Of the more than 190 adult population studies, relatively few studies seemed to contain random or representative samples of adult populations. The majority were based on university or college student populations (>60%), or with students together with other groups (an additional 5%). Otherwise, participants were mainly recruited through advertisements, postings on websites (e.g., Mechanical Turk), mailing lists, or personal appeal, or were carried out in specific work places or health care units. Some papers lacked a description of the selection process of study participants altogether. The number of study participants varied from 40 to 120,115. Studies were performed on all continents.

### 3.2. Measurements

The vast majority of the studies were based on self-reported exposures and outcomes, mostly through pen-and-pencil or web questionnaires, but sometimes also through telephone or face-to-face interviews. For younger children, parental reports about the child’s mobile phone use and health outcomes were used. The quantity of mobile phone use was mainly given in frequency and duration of calls and text messaging. However, with an increase of studies about smartphone usage, frequency and time spent on different apps and functions, including general screen time, were also examined. Many studies also included, for example, the type of phone, number of phones, from what age one had used a mobile phone, presence of a phone in the bedroom, what time the phone was used (e.g., time slots over the day, evening/nighttime use), and the size of the phone bill. A majority of the studies included scales or measurements of excessive or problematic mobile phone use (dependency/addiction), discussed further below.

Twelve studies could be identified as using objective measures for the quantity of mobile phone use. Three studies (conducted in the same population) used operator data for a subgroup of the participants [84,237,248]. The remaining studies used an app that was installed on the participants’ phones to log usage [49,53,91,174,175,176,200,239,258]. Two studies included a procedure where participants responded to questions about activity, including mobile phone use, several times per day on a given signal [26,95]. 

Additional measurement methods for mental health variables included structured psychiatric interviews [49,126,177,196,197], actigraphy for sleep [83,205], and sleep diaries [5,83,144,205]. Two studies included magnetic resonance imaging of the participants’ brains [110,283]. Further measurement methods occurred (e.g., body composition measurements, pedometers for physical activity, etc.), but did not pertain to mental health or psychological outcomes. 

### 3.3. Main Research Findings 

This section presents summaries and examples of the main findings in the included papers. The results have been clustered into three sections: (a) frequency/duration of mobile phone use and mental health outcomes, (b) bedtime mobile phone use, and (c) problematic mobile phone use. The main findings of each section are summarized in Table 3, Table 4 and Table 5. Table 6 summarizes the psychological factors that were most commonly associated with mobile phone use (all aspects).

#### 3.3.1. Frequency/Duration of Mobile Phone Use and Mental Health Outcomes

Among the studies of children and adolescents, a longitudinal study with 126 US adolescents found that more time spent on mobile phone use at baseline was associated with increased depression, measured with Becks Depression Inventory for Primary care at the one-year follow-up, while controlling for baseline depression [26]. In another longitudinal study, adolescents who owned a smartphone compared to non-owners slept less and had more sleep problems at baseline. Following up after two years, there were no differences in sleep problems between smartphone owners, new owners, and non-owners, but those who had owned a smartphone since baseline, compared to those who still did not own a smartphone, had shorter sleep duration on weekdays [249]. Cross-sectional associations were seen between quantity of mobile phone use and depressive symptoms in a study with 2785 Japanese adolescents [113], a study with 1328 Spanish adolescents/young adults [244], and a study with 7292 Finnish adolescents [139]. Overall mobile phone use of more than 5 h per day among Japanese adolescents was not associated with depression after adjusting for confounders, while using the mobile phone for more than 2 h per day for social networking services or online chatting was [264]. In a large British study with 120,115 adolescents, smartphone use on the weekends was negatively associated with mental well-being, while the associations for weekday use was non-linear, in that only use above an extreme cut-off was negative for mental well-being [227]. In an Israeli study of 185 children, daily time spent on a smartphone was not associated with psychopathological outcomes [250]. Regarding sleep outcomes, a longitudinal study of Japanese adolescents found mobile phone use of 2 h per day to be associated with new insomnia onset at the two-year follow-up [274]. A cross-sectional German study with 7533 adolescents found associations between higher mobile phone use and sleep problems among the girls in the crude analysis, but these were not statistically significant when controlling for confounders [149]. In a study with 6247 Chinese schoolchildren, time spent on texting, playing games, or surfing the internet on the mobile phone was associated with later bedtimes, shorter sleep duration, difficulties initiating and maintaining sleep, and daytime tiredness [117]. Time spent on the mobile phone was associated with shorter sleep duration and tiredness also among Japanese adolescents [113], and with poor sleep quality and daytime sleepiness in adolescents in Hong Kong [187]. In a Finnish study, mobile phone use was associated with deteriorated sleep habits and daytime tiredness in 12–14 years old girls and boys, and in 16–18 years old girls [228]. 

Among the studies on adult populations, a prospective study with 1127 Swedish university students found frequent mobile phone use at baseline to be a risk factor for sleep problems and depressive symptoms at the one-year follow-up in the men, and prolonged stress in the women [268]. This study, however, did not account for any confounding factors. Another prospective cohort study with 4159 Swedish young adults which, besides sex, accounted for educational level, occupation, and relationship status, showed similar results: Frequent mobile phone use was a risk factor for new cases of sleep problems in men, and for depressive symptoms in both men and women at the one-year follow-up [269]. Among the cross-sectional studies, frequency and duration of mobile phone use, logged by an app on the participants’ phones, was associated with depressed mood [239]. In another app log study, smartphone screen time was associated with depressed mood, but only before adjusting for confounders [53]. Cross-sectional associations were further seen between the frequency of calls and texts and perceived stress, sleep problems, and depressive symptoms among Swedish young adults [269]. A study that focused on work-related mobile phone use found that intensive mobile phone use among employees who had been provided by with a smartphone by the employer was associated with more work–home interference, less relaxation, less psychological detachment from work, and more exhaustion [65]. In other studies, time spent on the mobile phone was associated with anxiety [162], while the number of texts was associated with anxiety [29,162] and depressed mood [29]. A Finnish study with 6121 working-age participants, which examined mental symptoms in relation to the use of new technology, found associations between mobile phone use and depression in females 51–60 years, only [140]. Furthermore, in a US study with 308 adults, smartphone use frequency was negatively associated with depressive symptoms [74,75], and a Chinese study with 514 adults found that higher mobile use for calls was associated with higher mental well-being and positive affect [37]. 

Regarding personality, in one study, in which an app registered incoming and outgoing calls and text messages over five weeks among 49 German university students, associations between the number of calls and extraversion were seen, while no clear associations were found for the number of text messages and personality variables [200]. Another app log study found that smartphone use for calls was negatively associated with social anxiousness and loneliness [91]. One study concluded that lonely persons preferred to make voice calls rather than text messaging, while socially anxious persons preferred to text [231]. In a longitudinal study, increased mobile phone use over time was associated with decreased self-esteem and coping ability [286]. However, a one-week diary study that measured modes of social interaction found that meaningful text-based communication had a positive effect on self-esteem, compared to face-to-face communication and mobile phone voice communication [95]. Other studies found associations between time spent on mobile calls and extraversion [34] and low agreeableness [34,73], while text messaging was associated with neuroticism [34,73], extraversion [34], low self-esteem [73], low agreeableness [34], and low conscientiousness [34]. Time spent on mobile game playing was associated with low agreeableness [220,253].

#### 3.3.2. Bedtime Mobile Phone Use 

At least 35 studies addressed mobile phone use in the evening or at night: i.e., prior to bedtime, in bed, after “lights out”, awakening at night because of the phone, or even just the presence of a phone in the bedroom. About two thirds of these studies were based on children or adolescent populations.

A longitudinal Australian study that included 1101 adolescents found cross-sectional associations between nighttime phone use, poor sleep behavior, and depressed mood [286], but in longitudinal analyses, changes in nighttime phone use was not directly associated with subsequent changes in depressed mood. However, changes in sleep behaviors acted as a mediator between night-time phone use and subsequent depressed mood [286]. Another longitudinal study found cross-sectional associations between nighttime awakenings by the phone and sleep problems, perceived stress, and depressive symptoms in young adults, but no statistically significant prospective associations were seen at the one-year follow-up [269]. A diary study of work-related smartphone use at night showed subsequent lower sleep quantity, which in turn was associated with greater fatigue the next morning and less engagement during the work day [148].

In cross-sectional studies with children, as well as with adults, bedtime mobile phone use (in the broad definition, above) was associated with later bedtimes [16,22,31,82,85,88,93,223,263], longer sleep onset latency [53,79,112,223,293], shorter sleep duration [14,15,22,36,71,82,86,148,161,202,210], insomnia or sleep problems [5,14,79,85,97,144,199,202,205,235,269,293], reduced sleep quality or sleep efficiency [5,32,53,71,79,82,83,167,202,205], and reduced daytime functioning or tiredness [79,86,93,112,121,202,223,242,248,277,293]. In one study, keeping the phone close, rather than placing the phone at a distance from the bed, was associated with less sleep problems [235].

Almost all of the referred studies used self-reported sleep outcomes. However, two studies examined sleep by actigraphy in relation to self-reported mobile phone use [83,205]. Receiving night-time notifications on the phone predicted global sleep problems, subjective poor sleep quality, and sleep disruptions [205], and media use in bed or being awakened by the mobile phone at night negatively affected sleep efficiency [83]. 

Besides sleep outcomes, “bedtime” mobile phone use was associated with reduced mental health, suicidal feelings and self-injury [210], depressive symptoms [161,242,269,286], anxiety and stress [242], low self-esteem [286], and reduced cognitive performance in one study [235], but not in another [248]. 

#### 3.3.3. Problematic Mobile Phone Use 

Approximately 70% of the papers in this literature review addressed what can be termed “excessive” or “problematic” mobile phone use. They explored health outcomes of excessive mobile phone use, predictors for excessive use, such as personality or other psychological factors, or were reliability and validity studies of scales. Research about overuse, excessive, dependent, addictive, problematic, or pathological mobile phone use has emerged in parallel with the increased mobile phone usage. The constructs are commonly referred to as behavioral addictions and are likened with other non-substance addictions such as gambling addiction. As such, it seems to be a case of impaired ability to regulate one’s mobile phone use and can be associated with general symptoms of dependency, such as tolerance, withdrawal, escape, craving, using the mobile phone even when it is unsafe or prohibited, or functional consequences, such as financial or relational problems [295] (review, not included). A relationship can be seen with the concept of internet addiction, which was proposed as a specific psychiatric disorder in the 1990s by Young [296], who applied Diagnostic and Statistical Manual of Mental Disorders (DSM)-criteria for pathological gambling to internet use. Other constructs that have emerged include nomophobia and phubbing. Nomophobia is an abbreviation of “no mobile phone phobia” and refers to a phobia of not having access to a mobile phone [297]. It includes four dimensions: not being able to communicate, losing connectedness, not being able to access information, and giving up convenience [298]. The term “phubbing” comes from merging the words “phone” and “snubbing” and refers to when an individual is looking at or attending to his or her phone while in a conversation with others [124]. Yet another construct is “ringxiety”, or “phantom ringing”, which refers to perceiving that the phone rings even when it does not [260]. 

Excessive or problematic mobile phone use is usually associated with a high quantity of mobile phone use, while a high quantity of use does not necessarily imply problematic use. One of the papers concluded that mobile dependency was better predicted by personality factors (such as low self-esteem and extraversion) than actual phone use [108]. In one-month log data from 79 engineering students in Taiwan, a logarithm that combined frequency, duration, and frequency trend over time successfully predicted “smartphone addiction” [174,175]. Non-use patterns also predicted smartphone addiction [176]. Among functions that have been associated with excessive or problematic use are playing games [21,39,49,59,110,116,178] and the use of social networking sites (SNS) [33,39,49,116,183,209,224,285,288]. Another log data study showed that dependent participants, besides games and SNS, also used the phone more for web surfing, shopping, and entertainment, and less for talking and texting, than non-dependent participants [49]. 

A whole array of scales (>50) were used for examining problematic use in the papers. The great number is partly due to the fact that some scales existed in several versions, and that different names for what appear to be the same scales occurred, perhaps due to translations between languages. Several of the scales follow diagnosis criteria from the International Statistical Classification of Diseases and Related Health Problems (ICD) or DSM for pathological gambling or substance dependence, and some scales are direct adaptations of Young’s Internet Addiction Test [296], applied to mobile phones. Two of the most commonly referred to scales were the Mobile Phone Problem Use Scale (MPPUS) [25] and the Smartphone Addiction Scale (SAS) [146]. The MMPUS contains 27 items inspired from the addiction literature and covers areas such as tolerance, withdrawal, escape, craving, and negative consequences, giving a global score of problem use [25]. The SAS contains 48 items in six subscales: daily-life disturbance, positive anticipation, withdrawal, cyberspace-oriented relationship, overuse, and tolerance [146]. Several shortened versions of the scales were also used. 

The prevalence of problematic mobile phone use varied greatly in the studies, which can be expected because the measures, definitions, and study populations varied. Most of the studies were cross-sectional. Among the exceptions was a longitudinal study with 1877 Korean adolescents that used three yearly measurements [123]. The study found bidirectional relationships between mobile phone addiction and depressive symptoms over time [123]; i.e., mobile phone addiction had an influence on depressive symptoms, and depressive symptoms influenced mobile phone addiction, over time. Another study in the same population showed that high mobile phone addiction was associated with an increase in incidence of poor sleep quality over time [156]. In a Swedish study, subjective overuse of the mobile phone at baseline was a prospective risk factor for sleep disturbances at the one-year follow-up in female young adults [269]. 

In addition, cross-sectional associations were seen between excessive or problematic use and depression [7,18,39,42,62,80,89,90,94,98,100,105,123,130,131,168,180,184,185,189,214,244,251,256,267,269,282,290]. Conversely, in four studies, depression was negatively associated with problematic use [50,57,74,75]. Furthermore, associations were seen with anxiety [7,39,42,50,62,67,68,74,75,76,80,89,100,108,115,135,157,180,184,189,198,214,245,267] (but, a negative association between text message dependency and anxiety in Reference [185]), sleep problems or insomnia [7,32,115,269], reduced sleep quality [38,39,62,80,110,195,240], shorter sleep duration [110,130,179,289], eveningness [64,229,273], stress [18,46,89,105,106,116,131,143,180,243,269,280,285], lower general mental wellbeing [20,23,76,80,127,237], PTSD [55,56], suicidal thoughts [131,282,289], impulsivity or less self-control [27,28,29,30,33,46,56,67,68,102,110,116,119,120,130,137,166,233,234,256,283,288,292], attention deficit hyperactivity disorder (ADHD)-symptoms [252], productivity loss at work [72], and perceived phantom ringing [142,260]. Moreover, problematic use was associated with other behavioral addictions (e.g., internet addiction [12,19,43,45,50,52,63,100,105,118,127,145,146,154,178,186,198,217,236,266], shopping addiction [12,118,188], gambling addiction [78,245], and general addiction proneness [126,245]). Two studies examined participants with magnetic resonance imaging; when comparing mobile phone dependent subjects with non-dependent participants, differences in white matter integrity of the brain were seen [110,283].

Regarding psychological factors, several cross-sectional studies found associations between problematic mobile phone use and loneliness [24,91,98,129,133,158,270,279]. A longitudinal study with 288 participants 13–40 years of age examined causal relations between problematic use, loneliness, face-to-face-interaction, and the need for social assurance [132]. It found that loneliness predicted problematic use, while problematic use did not predict loneliness at the follow-up after four months. However, the authors concluded that loneliness increases problematic use, which in turn reduces face-to-face interactions and thus does not gratify increased needs for social assurance, and consequently, this process eventually leads to increased loneliness [132]. Other studies found associations with, e.g., shyness or social anxiousness [24,58,91,102,159], extraversion [12,13,18,25,46,64,81,108,255,256,261], fear of missing out [52,74,153,209,287], neuroticism [13,46,73,81,90,111,142,147,198,218,261,294], less self-esteem [13,25,100,108,256,281,289,291], low agreeableness [12,147], less openness [12,111,147,218,261], less conscientiousness [13,34,92,111,142,147,169,170], alexithymia [89], and less self-efficacy [99]. 

## 4. Discussion

The literature search showed that there is a vast—and increasing—amount of studies that explore links between mobile phone usage and mental health from a psychological or behavioral point of view. A high quantity of mobile phone use was associated with a wide range of mental health outcomes, such as depressive symptoms and sleep problems, in both children and adults. A relatively large proportion of the studies examined mobile phone use in relation to sleep habits; mobile phone use at bedtime or at night was associated with, e.g., shorter sleep and reduced quality of sleep. A dominating research field was excessive or problematic use, i.e., where intense mobile phone use is described as a behavioral addiction and/or pathological. A large amount of instruments to measure excessive or problematic use occurred, and problematic use was associated with several adverse outcomes, such as depression, anxiety, and sleep problems.

Only a few percent of the included studies had any form of longitudinal design. Cross-sectional studies limit the possibilities to draw valid conclusions about causal directions of associations. The found associations may thus be due to reversed causality, i.e., the outcome is causing what seems to be the risk factor, or the associations may be bi-directional or caused by common confounding factors not accounted for. For example, most of the studies on bedtime phone use and sleep variables were cross-sectional. In a longitudinal study with Canadian students [299] (not in the review due to the fact that mobile phone use was not analyzed separately), it was sleep problems that predicted media use and not the opposite. The researchers concluded that young adults used digital media to deal with sleep problems. Moreover, a study with 844 Belgian adults [300] (also not in the review) concluded that media, including mobile phones, was commonly used as a sleep aid. 

Further, a majority of the papers were based on self-reporting, which implies that both exposures and outcomes may be subject to misclassification, recall difficulty, recall bias, and response-style bias. It is previously known that there is rather low agreement between self-reported mobile phone use for calling or texting compared to logged data (e.g., [301]), and this applies also to smartphone usage [297]. However, it seems that applications that log smartphone usage are becoming more available, and thus are increasingly used in research. 

Strikingly, many of the studies on adult populations were done on university students or self-selected participants. This compromises generalizability of the results. Another observation was that in many studies, the found associations, although statistically significant, were small. 

The current literature review focused on studies with mobile phone use as a specific entity. Broadening the search to include more general terms such as “screen time”, “media use”, “technology use”, or “social media”, would lead to a higher quantity of studies with results that probably could apply also to mobile phone usage. Several different technologies (such as computers, tablets, or other hand-held devices) are used for the same activities and in the same contexts, and results from studies that include other technologies are seen to show similar results. However, a broader definition of the exposure was outside the scope of this review. 

Intense or frequent mobile phone usage is seen to be associated with a broad array of mental health related symptoms, behaviors, and psychological factors. Plausible behavioral and/or psychological mechanisms for the associations can be found in the review, such as impact on sleep habits, dependency/addiction issues, and individual personality traits. The extent to which mobile phone use interferes with the restorative functions of sleep can, of course, contribute to deteriorated health. Besides sleep being postponed, replaced, or disturbed by messages or calls at night, it is also conceivable that quantity as well as content of use can generate higher levels of psychological stress and physiological arousal. Higher levels of arousal can have a negative impact on sleep and recovery [302] and in other ways contribute to stress and ill health. In addition, there are studies [303,304] (not in the review) pointing to the fact that blue light emitted from screens may have an impact on melatonin levels and thus affect sleep and wakefulness. 

It is also conceivable that the time spent on devices takes time from other activities and health-related behaviors, such as physical activity, supportive social interactions, or staying on task at work or school. In the current review, several relevant aspects were excluded in the literature search, for example, the impact of mobile phone use on attention, consequences for relationships, cyberbullying, cyber sexual behaviors, and physical health outcomes, all aspects likely to potentially have an impact on mental health. Furthermore, this report does not account for all factors analyzed in the included papers. 

This review was done to supplement a systematic review of the potential health effects of exposure to radiofrequency electromagnetic fields (RF-EMF) from mobile phones. In light of this, it can be noted that there are several psychological and behavioral aspects that should be taken into consideration when assessing studies that examine health effects with RF-EMF exposure as the hypothesis. This is especially true given that many of the studies with an RF-EMF-perspective measure the exposures in the same manner as studies taking a psychological or behavioral perspective, i.e., with self-report. 

## 5. Conclusions

Associations between mobile phone use and adverse mental health outcomes are found in studies that take a psychological or behavioral perspective on the exposure. However, more studies of high quality are needed—with longitudinal design, objective measurements, and well-defined study populations—in order to draw valid conclusions about mechanisms and causal directions of associations.

## Figures and Tables

**Table 1 ijerph-15-02692-t001:** Search strategies in PubMed and PsycINFO 2018-03-19.

Database	Search Strings
PubMed	“cell phones” [MeSH Terms] OR “mobile phone” [Text Word] OR “mobile telephone” [Text Word] OR “cell phone” [Text Word] OR “cellular phone” [Text Word] OR “cellular telephone” [Text Word] OR “mobile phones” [Text Word] OR “mobile telephones” [Text Word] OR “cellular phones” [Text Word] OR “cellular telephones” [Text Word] OR smartphone [MeSH Terms] OR smartphone [Text Word] AND Stress [Title/Abstract] OR Depress* [Title/Abstract] OR Sleep* [Title/Abstract] OR Addict* [Title/Abstract] OR problem* [Title/Abstract] OR Mental [Title/Abstract] OR psychol* [Title/Abstract] OR psychi* [Title/Abstract] OR insomnia [Title/Abstract] OR compuls* [Title/Abstract] OR patholog* [Title/Abstract] OR dependen* [Title/Abstract] OR anxi* [Title/Abstract] OR symptom* [Title/Abstract] AND (“1993/01/01” [PDat]: “2017/12/31” [PDat]) AND (English [lang] OR Norwegian [lang] OR Swedish [lang])
PsycINFO	ti (“cell phones” OR “mobile phone” OR “mobile telephone” OR “cell phone” OR “cellular phone” OR “cellular telephone” OR “mobile phones” OR “mobile telephones” OR “cellular phones” OR “cellular telephones” OR “smart phone” OR “smart phones” OR “smartphone” OR “smartphones”) OR ab(“cell phones” OR “mobile phone” OR “mobile telephone” OR “cell phone” OR “cellular phone” OR “cellular telephone” OR “mobile phones” OR “mobile telephones” OR “cellular phones” OR “cellular telephones” OR “smart phone” OR “smart phones” OR “smartphone” OR “smartphones”) AND ti (Stress OR Depress* OR Sleep* OR Addict* OR problem* OR Mental OR psychol* OR psychi* OR insomnia OR compuls* OR patholog* OR dependen* OR anxi* OR symptom*) OR ab(Stress OR Depress* OR Sleep* OR Addict* OR problem* OR Mental OR psychol* OR psychi* OR insomnia OR compuls* OR patholog* OR dependen* OR anxi* OR symptom*) Filter Peer review, Eng, No 1993/01/01–2017/12/31

**Table 2 ijerph-15-02692-t002:** Number of included papers (*n* = 290) by publication year.

2001	2002	2003	2004	2005	2006	2007	2008	2009	2010	2011	2012	2013	2014	2015	2016	2017 ^1^
1	1	1	-	1	3	7	4	7	6	7	8	17	37	46	59	85

^1^ Six papers were dated 2018 but had been published online previously and were categorized as 2017.

**Table 3 ijerph-15-02692-t003:** Frequency/duration of mobile phone use: summary of main results.

Outcomes	Study Designs and Citations
• Depression	L: [26,268,269] CS: [29,53,113,139,140,239,244,264,269] NA: [74,75]
• Sleep problems, lower sleep quality	L: [268,269,274] CS: [117,149,187,269]
• Later bedtimes, shorter sleep	CS: [103,113,117,228]
• Tiredness, reduced daytime function	CS: [65,113,117,187,228]
• Lower mental well-being	CS: [227] NA: [37]
• Stress	L: [268] CS: [269]
• Anxiety	CS: [29,162]

L = Longitudinal, CS = Cross-sectional, NA = Negative association. In crude, but not in adjusted, analyses: reference 53, 149. In subgroup of older women: reference 140.

**Table 4 ijerph-15-02692-t004:** Bedtime mobile phone use: summary of main results.

Outcomes	Study Designs and Citations
• Sleep problems	CS: [5,14,79,85,97,144,199,202,205,235,269,293]
• Lower sleep quality/efficiency	CS: [5,32,53,71,79,82,83,167,202,205]
• Longer sleep onset latency	CS: [53,79,112,223,293]
• Poor sleep behavior	L: [286] CS: [286]
• Later bedtimes	CS: [16,22,31,82,85,88,93,223,263]
• Shorter sleep	CS: [14,15,22,36,71,82,86,148,161,202,210]
• Tiredness, reduced daytime function	L: [148] CS: [79,86,93,112,121,202,223,235,242,248,277,293]
• Depression	CS: [161,210,242,269,286]

L = Longitudinal, CS = Cross-sectional.

**Table 5 ijerph-15-02692-t005:** Problematic mobile phone use: summary of main results.

Outcomes	Study Designs and Citations
• Depression	L: [123] (bidirectional) CS: [7,18,39,42,62,80,89,90,94,98,100,105,123,130,131,168,180,184,185,189,214,244,251,256,267,269,282,290] NA: [50,57,74,75]
• Anxiety	CS: [7,39,42,50,62,67,68,74,75,76,80,89,100,108,115,135,157,180,184,189,198,214,245,267] NA: [185]
• Sleep problems	L: [269] CS: [7,32,115,269]
• Lower sleep quality	L: [156] CS: [38,39,62,80,110,195,240]
• Shorter sleep	CS: [110,130,179,289]
• Stress	CS: [18,46,89,105,106,116,131,143,180,243,269,280,285]
• Lower mental wellbeing	CS: [20,23,76,80,127,237]
• Other behavioral addictions	CS: [12,19,43,45,50,52,63,78,100,105,118,126,127,145,146,154,178,186,188,198,217,236,245,266]

L = Longitudinal, CS = Cross-sectional, NA = Negative association.

**Table 6 ijerph-15-02692-t006:** Summary of the psychological factors most commonly associated with mobile phone use (all aspects).

Psychological Factors	Citations
• Impulsivity/less self-control	[27,28,29,30,33,46,56,67,68,102,110,114,116,119,120,130,137,166,233,234,256,283,288,292]
• Extraversion	[12,13,18,25,34,46,64,81,108,200,255,256,261]
• Neuroticism	[13,34,46,73,81,90,111,142,147,198,218,261,294]
• Less self-esteem	[13,25,73,100,108,256,281,286,289,291] NA: [95]
• Loneliness	[24,91,98,129,132,133,158,231,270,279] NA: [91]
• Less conscientiousness	[13,34,92,111,142,147,169,170]
• Low agreeableness	[12,34,73,147,220,253]
• Social anxiety, shyness	[24,58,91,102,159,231] NA: [91]
• Less openness	[12,111,147,218,261]
• Fear of missing out	[52,74,153,209,287]

NA = Negative association.

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
