# Peer review of "Mobile Phone Use and Mental Health. A Review of the Research That Takes a Psychological Perspective on Exposure"

_ijerph, 2018, doi:10.3390/ijerph15122692_

Round 1

Reviewer 1 Report

Although it is a narrative review, I would like to see some summary statistics in Tables or Figures for the hurried reader. It also seems that this would be a good basis for a more formal meta-analysis to extract effects sizes.

Line 77: please avoid “approximately 10” and report the exact number.

Lin174: please add higher or lower to extraversion, I assume higher; also I assume with high neuroticism

Author Response

Thank you for considering the manuscript for publication in IJERPH! I am grateful to the two reviewers for giving relevant and valuable comments and suggestions. I have carefully considered these and revised the manuscript accordingly. See my point by point responses (R) below.

All changes can be found through “tracked changes”. Besides those based on the suggestions from the two reviewers, minor editing has been done throughout. I have inserted 4 tables. I have also updated the reference list and re-inserted Appendix A (Prisma Diagram) as there were flaws in the previous version. In addition, the Appendix A is submitted as a separate file, in case the editorial office has a preference in which format it should be inserted.

Response to Reviewer 1.

1.       Although it is a narrative review, I would like to see some summary statistics in Tables or Figures for the hurried reader. It also seems that this would be a good basis for a more formal meta-analysis to extract effects sizes.

R: Thank you for this suggestion. I have added 3 tables (Tables 3-5) to summarize the main results of the three main aspects of mobile phone use. They give coarse descriptions of outcomes and study designs. I have also added a table (Table 6) to describe the main personality factors found to associate with mobile phone use. Because of the great number of included studies, effect sizes are not presented. Yet, I hope the tables should help the reader to identify the main trends of the research.

2.       Line 77: please avoid “approximately 10” and report the exact number.

R: Approximately 10 is now omitted.

3.       Line 174: please add higher or lower to extraversion, I assume higher; also I assume with high neuroticism

R: Instead of adding the adjectives, I changed the order of the presented personality factors, and hope this sufficiently addresses the issue. 

Reviewer 2 Report

This is an interesting study and presents a much-needed account of how mobile use may be related to mental health related issues. The author has undertaken a useful review and drawn out a number of key issues and valid conclusions from these. I have some minor observations which I feel can easily be addressed by the author but overall am satisfied that this paper is suitable for publication:

#1. The results suggest that the majority of studies are based on self reported exposures and outcomes (as noted by the author on p3), therefore, perhaps a more representative title of the paper is in reference to “a psychological perspective” (rather than behavioural perspective), given that there is little available evidence which actually uses behavioural data (although it is noted 12 studies use objectives measures to quantify mobile phone use but these are a minority).

#2. Some of the opening statements would benefit from some citations to substantiate the claims. E.g., “This fast development with changed exposure patterns has raised questions about potential health effects of the exposure”. Please include some citations here to those who have raised these concerns.

#3. P2- it is stated that qualitative, case or experimental studies were excluded (amongst other things). It is not clear what the rationale is for this. That is, what judgement or criteria is being used to assume that qualitative studies are not appropriate as evidence for this area of research? This then corresponds to point on p3 which states “The identified studies (n = 290) mainly dealt with quantitative mobile phone use…”. Well, they would all be quantitative if they were screened in this way to remove qualitative ones! I think readers would like greater justification for how exclusion criteria were determined and the extent to which a qualitative study which is self reporting is considered inferior to the majority of the papers included which are also self report (and even in some cases include phone interviews as the methodological approach).

Author Response

Thank you for considering the manuscript for publication in IJERPH! I am grateful to the two reviewers for giving relevant and valuable comments and suggestions. I have carefully considered these and revised the manuscript accordingly. See my point by point responses (R) below.

All changes can be found through “tracked changes”. Besides those based on the suggestions from the two reviewers, minor editing has been done throughout. I have inserted 4 tables. I have also updated the reference list and re-inserted Appendix A (Prisma Diagram) as there were flaws in the previous version. In addition, the Appendix A is submitted as a separate file, in case the editorial office has a preference in which format it should be inserted.

Response to Reviewer 2.

This is an interesting study and presents a much-needed account of how mobile use may be related to mental health related issues. The author has undertaken a useful review and drawn out a number of key issues and valid conclusions from these. I have some minor observations which I feel can easily be addressed by the author but overall am satisfied that this paper is suitable for publication:

1.       The results suggest that the majority of studies are based on self reported exposures and outcomes (as noted by the author on p3), therefore, perhaps a more representative title of the paper is in reference to “a psychological perspective” (rather than behavioural perspective), given that there is little available evidence which actually uses behavioural data (although it is noted 12 studies use objectives measures to quantify mobile phone use but these are a minority)

R: Thank you for this suggestion! I agree and have changed to “psychological” in the title, but have kept “psychological and behavioral” in several places in the text.

2.       Some of the opening statements would benefit from some citations to substantiate the claims. E.g., “This fast development with changed exposure patterns has raised questions about potential health effects of the exposure”. Please include some citations here to those who have raised these concerns.

R: Citations have been added (WHO and USFDA)

3.       P2- it is stated that qualitative, case or experimental studies were excluded (amongst other things). It is not clear what the rationale is for this. That is, what judgement or criteria is being used to assume that qualitative studies are not appropriate as evidence for this area of research? This then corresponds to point on p3 which states “The identified studies (n = 290) mainly dealt with quantitative mobile phone use…”. Well, they would all be quantitative if they were screened in this way to remove qualitative ones! I think readers would like greater justification for how exclusion criteria were determined and the extent to which a qualitative study which is self reporting is considered inferior to the majority of the papers included which are also self report (and even in some cases include phone interviews as the methodological approach).

R: Thank you for pointing out some unclarities in the manuscript! The aim was to review quantitative observational studies (in a similar fashion to the procedures of the WHO expert group but focusing on psychological mechanisms rather than electromagnetic fields). I have clarified by adding "quantitative" on lines 41 and 50. 

I do not feel that qualitative studies are inferior to quantitative studies, especially when it comes to identifying underlying psychological mechanisms for associations or exploring new grounds. But with the aim to review quantitative studies, qualitative studies were excluded. Actually, not so many qualitative studies emerged in the search! 

To go on, I see that the first sentence on P3 Results was confusing. I have rewritten the first paragraph to clarify the three main areas of the included papers.

I agree that self-report not necessarily is inferior to objective reports. But if we want to do health risk assessments concerning frequency/duration of mobile phone use, it is a good idea to supplement self-report studies with studies that also use objective measures. Self-reported mobile phone use is subject to several biases, such as recall bias or response-style bias (e.g., mood may affect estimation of how much the phone is used). Correctly estimating frequency and duration of mobile phone use, e.g., the past week, is difficult.

I have changed the wording “good quality” to “high quality” in the conclusions (to tone down that all studies that aren't longitudinal or use objective measures would be "not-good").